# Urinary DNA as a Tool for Germline and Somatic Mutation Detection in Castration-Resistant Prostate Cancer Patients

**DOI:** 10.3390/biomedicines11030761

**Published:** 2023-03-02

**Authors:** Tomas Januskevicius, Rasa Sabaliauskaite, Daiva Dabkeviciene, Ieva Vaicekauskaite, Ilona Kulikiene, Agne Sestokaite, Asta Vidrinskaite, Arnas Bakavicius, Feliksas Jankevicius, Albertas Ulys, Sonata Jarmalaite

**Affiliations:** 1Clinic of Gastroenterology, Nephro-Urology and Surgery, Institute of Clinical Medicine, Faculty of Medicine, Vilnius University, M. K. Ciurlionio st. 21/27, LT-03101 Vilnius, Lithuania; 2Laboratory of Genetic Diagnostic, National Cancer Institute, Santariskiu st. 1, LT-08406 Vilnius, Lithuania; 3Biobank, National Cancer Institute, Santariskiu st. 1, LT-08406 Vilnius, Lithuania; 4Division of Human Genome Research Centre, Institute of Biosciences, Life Sciences Center, Vilnius University, Sauletekio Ave. 7, LT-10257 Vilnius, Lithuania; 5Nuclear Medicine Department, National Cancer Institute, Santariskiu st. 1, LT-08660 Vilnius, Lithuania; 6Urology Centre, Vilnius University Hospital Santaros Klinikos, Santariskiu st. 2, LT-08661 Vilnius, Lithuania; 7Oncourology Department, National Cancer Institute, Santariskiu st. 1, LT-08660 Vilnius, Lithuania

**Keywords:** DNA damage repair, *BRCA1/BRCA2* genes, castration-resistant prostate cancer, abiraterone acetate

## Abstract

(1) Background: DNA damage response (DDR) pathway gene mutations are detectable in a significant number of patients with metastatic castration-resistant prostate cancer (mCRPC). The study aimed at identification of germline and/or somatic DDR mutations in blood and urine samples from patients with mCRPC for correlation with responses to entire sequence of systemic treatment and survival outcomes. (2) Methods: DDR gene mutations were assessed prospectively in DNA samples from leukocytes and urine sediments from 149 mCRPC patients using five-gene panel targeted sequencing. The impact of DDR status on progression-free survival, as well as treatment-specific and overall survival, was evaluated using Kaplan–Meier curves and Cox regression. (3) Results: DDR mutations were detected in 16.6% of urine and 15.4% of blood samples. *BRCA1*, *BRCA2, CHEK2*, *ATM* and *NBN* mutations were associated with significantly shorter PFS in response to conventional androgen deprivation therapy and first-line mCRPC therapy with abiraterone acetate. Additionally, *BRCA1* and *BRCA2* mutation-bearing patients had a significantly worse response to radium-223. However, DDR mutation status was predictive for the favourable effect of second-line abiraterone acetate after previous taxane-based chemotherapy. (4) Conclusions: Our data confirm the benefit of non-invasive urine-based genetic testing for timely identification of high-risk prostate cancer cases for treatment personalization.

## 1. Introduction

Prostate cancer (PCa), the second most frequent cancer in men, has a wide variety of clinical behaviours, ranging from latent to rapidly progressing disease [1]. Age, family history, racial and ethnic background are the main risk factors for PCa [2,3]. It is assumed that about 60% of individual variation in PCa risk can be attributed to genetic factors [4]. Recent data suggest that up to 17.2% of PCa patients may harbour germline mutations in non-androgen receptor-related pathways [5,6,7]. In PCa, heritable mutations predominantly occur in *BRCA1, BRCA2*, *ATM* and *CHEK2* genes responsible for DNA damage repair (DDR) pathway responses and the genomic integrity of cells. Inherited mutations in DDR genes increase the risk of a more aggressive form of PCa, and germline *BRCA1/BRCA2* mutations are confirmed biomarkers of poor prognosis [8]. Moreover, somatic DDR pathway mutations are frequent in tumour cells, especially in metastatic disease, making them an attractive target for personalized therapy [8].

Nowadays, germline testing of *BRCA1/BRCA2* and other DDR genes is recommended in the early stages of PCa by various guidelines, including the NCCN, ESMO and EAU, for heritable disease identification, while tumour (somatic) mutation testing is mainly advocated for metastatic castration-resistant PCa (mCRPC) in relation to poly (ADP-ribose) polymerase (PARP) inhibitor treatment selection [9,10]. Since tumour testing in mCRPC is not always feasible, liquid biopsies can serve as non-invasive alternative sources of diagnostic material.

Liquid biopsy is an innovative tumour-specific mutation detection tool that can be applied to various body fluids such as saliva, blood, urine, and other clinical biosamples. DNA originating from cell-free circulating tumour DNA (cfDNA) or circulating tumour cells (CTC) possesses the same mutation profile as primary tumour or metastases and can represent wide heterogeneity of cancer loci and clonal evolution leading to mCRPC [11]. Additionally, compared to tissue biopsies, liquid biopsy studies allow a more accurate assessment of molecular changes beyond the tumour, including the possibility of detecting both germline and somatic mutations. Urine, as a non-invasive form of liquid biopsy, is an ideal type of biosample: it can be collected frequently in large quantities without causing discomfort, thus ensuring early detection of relapse, resistance to the prescribed treatment or even metastasis. Larger amounts of prostate-specific cells and cfDNA can be collected after prostate massage, but due to the unpleasant procedure, it is increasingly replaced by the use of plain voided urine [12]. First-void urine can be easily separated by centrifugation into a pellet enriched in cells and proteins and supernatant containing cell-free DNA and exosomes.

The discovery and characterization of DDR mutations in PCa accelerated the development of novel personalised treatment options, and two PARP inhibitors, olaparib and rucaparib, were recently approved for systemic mCRPC treatment [13]. Accumulating data also suggest the impact of inherited DDR defects on PCa response to conventional therapies, such as next-generation hormonal therapy (abiraterone acetate (AA), enzalutamide) [14], radium-223 [15] and platinum-based chemotherapy [16,17,18], but such observations need further validation.

The present study aimed at evaluating the prevalence of germline and somatic mutations in five DDR pathway genes (*BRCA1*, *BRCA2*, *CHEK2*, *ATM*, and *NBN*) in blood and urine samples of mCRPC patients from a Lithuanian cohort. For predictive value validation, clinical response to a whole spectrum of systemic treatments, from androgen deprivation therapy in a hormone-sensitive setting to radium-223 in cases with bone metastasis, and survival outcomes were evaluated with respect to DDR mutation status.

## 2. Materials and Methods

### 2.1. Patient Cohort

In total, 149 consecutive patients with histologically confirmed mCRPC treated and followed at the National Cancer Institute between 2017 and 2018 were included in the study. mCRPC was defined as a metastatic PCa with castration level of serum testosterone (<1.7 nmol/L), as well as biochemical and/or radiological progression of the disease. The study was approved by the Regional Bioethics Committee (No.: 158200-17-874-411), and written informed consent was obtained from all participants.

All patients were followed with monthly clinical examination, PSA and alkaline phosphatase (ALP) assessment every 2 to 3 months, and radiological examination (bone-scan and body CT) at least every 6 months. Progression-free survival (PFS) after different types of therapy was defined as the time elapsed between treatment initiation and disease progression or death from any cause, whichever occurred first. Disease progression was confirmed when at least two of three progression criteria (PSA progression, radiographic progression and clinical deterioration) were fulfilled. Overall survival was defined as the time from PCa diagnosis to death from any cause.

### 2.2. Sample Collection and DNA Extraction

Blood and first-void urine samples were collected prospectively into EDTA blood collection tubes and urine collection containers according to the standardized procedures. Urine samples were processed within 30 min according to the following protocol: the cellular content of the urine sample was pelleted by centrifugation at 2000× *g* for 15 min, and the supernatant was removed. The cell pellets were washed with PBS and precipitated. Afterwards, the supernatant was removed, and the cell pellets were resuspended in 2 mL PBS and stored at −80 °C until use. Leukocyte DNA extraction was performed by a fully automated robotic QIAcube system workstation by using QIAamp DNA Blood Mini Kit (Qiagen, Hilden, Germany). Urine samples were processed within 30 min after the samples were taken, using Viral RNA Mini Kit (Qiagen, Hilden, Germany) according to manufacturer instructions. DNA concentration and purity were determined using the NanoDrop 2000 spectrophotometer (Thermo Scientific, Wilmington, DE, USA) and stored at −20 °C until use.

### 2.3. Quantitative PCR

For the detection of the predominant DNA response pathway gene mutations in genomic DNA, custom TaqMan™ SNP Genotyping Assays (Applied Biosystems (AB) Thermo Fisher Scientific (TFS), Paisley, UK) were designed: 6 for *BRCA1*/*BRCA2* (rs80359112, rs80356898, rs397507246, rs80357711, rs28897672A_C, rs80359604_GT), three for *CHEK2* (rs555607708, rs17879961, rs121908698) and one for *NBN* (rs587776650) gene. Quantitative PCR (qPCR) reactions were performed according to the manufacturer’s protocol. All qPCR reactions were performed in duplicates following the manufacturer’s protocol and using a 7500 Real-Time PCR System (AB, TFS, Foster, CA, USA).

### 2.4. Targeted Next-Generation Sequencing

Germline DNA from DDR mutation-negative cases in qPCR screening and all urine samples (*n* = 133 and *n* = 139, respectively) was sequenced using a targeted five-gene panel: *BRCA1*, *BRCA2*, *CHEK2*, *ATM* and *NBN*. The *BRCA* Germline I Reference Standard gDNA (Horizon Discovery, Cambridge, UK) was used as a positive control. Genomic DNA concentration was determined by using Qubit™ dsDNA BR Assay Kit on a Qubit™ 2.0 Fluorimeter (Invitrogen, TFS, Eugene, OR, USA). For library preparation, Ion AmpliSeq™ Library Kit 2.0 and custom On-Demand Panel (from Life Technologies (LT), Carlsbad, CA, USA) were used under conditions provided by the manufacturer’s protocol. Library concentrations were determined by using the Ion Library TaqMan™ Quantification Kit (AB, TFS, Vilnius, Lithuania). Sequencing was performed on the Ion Torrent™ Ion S5™ system. Average read length after sequencing was 209 bp: ≥93% on target and ≥98% uniformity. Data analysis was performed with Ion Reporter 5.10 tool (LT, Carlsbad, CA, USA). Sequence reads were aligned to human reference genome—hg19. The frequency of pathogenic and likely pathogenic mutations was estimated for each gene and confirmed according to the ClinVar database also using the Integrative Genomics Viewer 2.4.8 tool to eliminate false positive cases and to confirm detected mutant reads in both sequence directions.

### 2.5. Statistical Analysis

Associations between categorical variables were evaluated by using two-sided Chi-square test or Fisher’s exact test, as appropriate. Normal distribution was tested using Shapiro–Wilk W test. Two independent samples were analysed by Mann–Whitney U test. Outliers were defined as cases exceeding three interquartile ranges for PFS—study patients that discontinued regular follow-up protocol—and removed from the analysis. Kaplan–Meier curves and Cox regression were used for survival analyses. Regarding multivariate Cox regression, covariates with p levels up to 0.2 were selected for the analysis. Differences were considered statistically significant when *p*-value was <0.050. The data were analysed using R x64 4.0.3 (R Foundation for Statistical Computing, Vienna, Austria), RStudio 1.4.1717 (Posit, PBC, Boston, MA, USA) and IBM SPSS Statistics 21 (IBM Corp., Armonk, NY, USA) software.

## 3. Results

### 3.1. DDR Mutations in Blood and Urine of mCRPC Patients

In a cohort of 149 mCRPC cases, 23 patients (15.4%) were identified with germline mutations of selected DDR genes in the blood cells: 16 with qPCR and seven using NGS. According to mutation type, six insertion/deletion and 17 missense mutations were detected. In the blood samples from the patients, *BRCA1* mutation was detected in two (1.3%), *BRCA2* in three (2.0%), *CHEK2* in 12 (8.1%), *ATM* in five (3.4%), and *NBN* in one (0.7%). The most frequently mutated gene was *CHEK2*: 11 patients harboured *CHEK2* mutation *c.470T > C* and one had the *c.1100delC* mutation. Three variants of unclear clinical significance according to the ClinVar database were detected (Table 1) in leukocytes.

For the mutation analysis, urine samples from 139 mCRPC patients were available (six samples were missing and four were low in NGS library quantity). In total, 26 pathogenic mutations were detected in 23 out of 139 cases (16.6%; Figure 1). All germline mutations detected in leukocytes (19 in total) from these patients were also identified in urine samples. However, seven somatic mutations (2—*BRCA1*, 2—*BRCA2*, 1—*CHEK2* and 2—*ATM*) were additionally identified in the urine of five patients (5/139; 3.6%), where the majority (116/139; 83%) were negative for germline alterations in the blood analysis.

Multiple mutations were detected in the urine of three patients (2.2%; 3/139), where concomitant somatic alterations in *BRCA1* and *ATM* genes were detected in one case (0.7%; 1/139) and a combination of *BRCA2* and *CHEK2* mutations was identified in two patients (1.4%; 2/139; somatic-somatic and somatic-germline).

### 3.2. Urinary DDR Mutations and Clinical Response

Due to the more complete presentation of both somatic and germline types of mutations, an oncoprint of urinary samples was used for further analysis. The patients with and without detected pathogenic mutations in urine were divided into mutation-positive DDR(+) and mutation-negative DDR(−) groups, respectively. Although there were 23 cases with at least one mutation detected in urine, two cases (PN029, PN038) were classified as outliers and not included in the analysis. Additionally, one patient (PN072) was evaluated as DDR(−) due to the unknown pathogenic variant of *ATM* mutation (Table 1). Finally, the patients with *BRCA1, BRCA2*, *CHEK2* (c.1100delC only), *ATM* and *NBN* were classified as DDR(+)A group (12 patients), omitting the cases with less pathogenic [4] *CHEK2* mutation c.470T > C (eight patients). Therefore, the analysis was performed by comparing the DDR(+) with the DDR(−) group, and the DDR(+)A with the DDR(−) group. Baseline clinical and demographic characteristics of DDR(+) and DDR(−) cohorts are provided in Table 2.

Mean response time to androgen deprivation therapy (ADT) in a hormone-sensitive setting (HSPC) was 4.5 (95% CI: 3.9–5.1) years. For DDR(+)A patients, the mean response time was 1.8 times shorter, as compared to DDR(−) patients (Figure 2B, *p* = 0.022). In multivariate regression analysis, mutations of the DDR(+)A cohort (*p* = 0.019) and higher cISUP grade group (*p* = 0.001) were the main predictors for the shorter response to conventional ADT (Table 3).

After disease progression to mCRPC, the mean PFS for the first-line therapy was 1.3 (95% CI: 1.1–1.5) years. DDR(+) was associated with significantly shorter PFS (Figure 2C, *p* = 0.003). In addition, significantly worse outcomes were registered in the DDR(+)A group vs. DDR(−) (Figure 2D, *p* = 0.005). The same response was evaluated in multifactorial analysis (Table 3).

Regarding AA as the first-line drug for mCRPC, DDR(+) and DDR(+)A groups were associated with significantly shorter response times, as compared to DDR(−) patients (Figure 2E–F, *p* = 0.003 and *p* = 0.025, respectively). Likewise, adjusted HR was significantly higher for DDR(+) vs. DDR(−) (Table 3, *p* = 0.011), when a tendency was demonstrated by the DDR(+)A mutation group (Table 3, *p* = 0.076). No statistically significant differences in PFS for the first-line therapy were observed when docetaxel was administered as the first-line drug.

Eight DDR(+) and 23 DDR(−) patients received AA after disease progression on prior docetaxel therapy. The mean PFS for the AA as the second-line drug was 1.5 (95% CI: 0.9–2.1) years, while mean PFS for DDR(+) and DDR(−) patients was 1.7 (95% CI: 0.8–2.6) and 1.4 (95% CI: 0.8–2.1) years, respectively. Contrary to the first-line treatment with AA, DDR(+) status was not a significant predictor of poor response to the second-line treatment with AA (HR = 0.9 95% CI: 0.4–2.0, *p* = 0.78).

Radium-223 dichloride therapy was administered to 16 patients in the DDR(−) group and seven patients in the DDR(+) group. Regarding DDR(+) status, two patients with two *CHEK2* mutations and five patients with *BRCA1/BRCA2* mutations underwent radium-223 dichloride therapy. After three or six cycles of bone-specific therapy, the median levels of ALP were comparable in all groups with particular DDR status. However, the DDR(+)A group showed a tendency toward a worse response on bone scintigraphy at the end of radium-223 therapy than the DDR(−) group (*p* = 0.11). In more detail, only two of five DDR(+)A patients showed a positive response, while 81% (13 of 16) of DDR(−) and 57% (4 of 7) of DDR(+) patients had a positive response.

### 3.3. Survival and Urinary DDR Mutations

The mean overall survival for mCRPC patients was 11.2 (95% CI: 10.2–12.2) years, where the DDR(+)A group was characterized by reduced survival rates, as the mean survival was 9.5 (95% CI: 7.1–11.9) vs. 11.4 (95% CI: 10.3–12.4) in the DDR(−) group (Figure 2H, *p* = 0.05). Although multivariate analysis identified the cISUP grade group and radical therapy as independent prognostic factors for overall survival (Table 3, *p* < 0.001 and *p* = 0.001, respectively), gene alterations in the DDR(+)A group (HR = 2.0, 95% CI: 1.0–4.2, *p* = 0.058) also demonstrated a remarkably deleterious impact on overall survival in mCRPC patients (Table 3).

## 4. Discussion

CRPC is a clinically heterogenous disease with various treatment possibilities approved for the therapeutic armamentarium. However, for the selection of the most appropriate treatment strategies and further drug sequencing, scientific evidence is still scarce, especially in the most aggressive and lethal form of the disease—mCRPC, including DDR-mutated cases.

According to the recent literature, the percentage of patients with germline mutations in DDR genes ranged from 1.4–2.1% in low-risk [7,18] to 11.8% in metastatic [7] disease, while it could be as high as 15.4% in mCRPC patients, as detected in our study. In comparison, the median prevalence rates of somatic mutations vary from 10.7% in PCa in general to 13.2% in the metastatic disease [19]. In our study, 16,6% cases were identified with germline or somatic DDR pathway mutations using urine samples as liquid biopsies.

Pathogenic germline *BRCA1*, *BRCA2* and *ATM* mutations in blood leukocytes were detected in 3.4% of mCRPC patients each, while the *CHEK2* mutation was the most common (8.1%), though *CHEK2 c.1100delC* was detected in only 0.7% of patients. Other authors have reported comparable rates, where *BRCA1*, *BRCA2* and *ATM* mutations were detected in 4.1% and 1.9% of mCRPC patients, respectively [19]. Importantly, all inherited DDR mutations detected in blood samples were also identified in urine, where a significant number of additional somatic alterations were also detected.

The high diagnostic potential of liquid biopsy tests has been previously shown in a blood plasma study by Annala et al., where 65 mCRPC patient germline and cfDNA sample pair sequencing showed high concordance in allele fractions and somatic mutation status [20]. Moreover, liquid biopsy is more suitable for patient monitoring, metastasis detection, and disease heterogeneity analysis, since multiple prostate biopsies are not recommended for mCRPC patients [21]. While the mutation analysis in urine already has been suggested as a diagnostic test for PCa [22], comprehensive data are lacking on the amount of germline and/or somatic mutations in urine of mCRPC cases. In our study, in addition to germline mutations detected both in blood leukocytes and urine DNA, an additional seven somatic mutations were identified on NGS in urine samples.

It is generally accepted that PCa patients harbouring DDR mutations have a more aggressive form of the disease and are associated with poor survival outcomes [23,24]. Emerging evidence suggests that DDR mutation carriers demonstrate an inferior response to androgen pathway-targeted therapy [25], while based on the synthetic lethality phenomenon, they may benefit from poly-ADP ribose polymerase (PARP) inhibitors [26,27,28]. From this point of view, patients with DDR mutations may also obtain a greater clinical benefit from DNA-damaging therapies, such as platinum-based chemotherapy [16,17,18] or bone-targeted radium-223 therapy [15,29]. Regrettably, these studies analysing clinical response in DDR mutated PCa cases typically cover only some fragments of the disease with specific clinical characteristics and provide hardly comparable controversial results.

CTC and cfDNA continue to be extensively researched in the liquid biopsy field. Identification of total CTC and other CTC-derived biomarkers, such as androgen receptor splice variant 7 (AR-V 7), in advanced PCa patients, might predict the disease prognosis and treatment response, as shown by various studies [30,31,32]. Further investigation into DDR mutation-positive PCa and CTC-related biomarkers may lead to a better understanding of the disease resistance and the development of personalized treatment options.

Our study provides a comprehensive analysis of clinical responses to various PCa therapies in all stages of the disease, from ADT in HSPC to bone-targeted therapy in mCRPC, and represents a complete natural history of disease progression for DDR mutation carriers. According to our data, DDR(+)A mutation carriers demonstrated significantly shorter response to conventional ADT in HSPC that consequently was followed by significantly shorter PFS for AA in the mCRPC setting, and similar results were also reported by Annala et al. [25]. All of these findings, including relatively rapid disease progression in DDR mutated cases, provide a rationale for germline and somatic mutation testing for timely identification of the most aggressive forms of PCa. Moreover, our study demonstrated a superior response of DDR mutation carriers to AA when it was administered after docetaxel, while the first-line treatment with docetaxel also showed sufficient response in these patients. Consistent with our finding, Castro et al. reported comparable response rates between patients with and without DDR mutations treated with taxane therapy [19]. A favourable response to platinum-based chemotherapy in DDR mutation carriers with mCRPC is also known from other studies [16,17,18], supporting the idea that targeting different cellular pathways (cell cycle, apoptosis, DNA repair) might be beneficial for improved response in PCa patients with DDR deficiency. In fact, DDR(+)A patients responded poorly to a set of administered therapies, and multivariate analysis revealed DDR(+)A mutations as remarkably hazardous to overall survival after adjustment to cISUP grade and radical therapy. Shorter time to progression in various treatment regimens observed for DDR(+) patients in our and previous studies [15,25] suggest the need to review the whole sequence of systemic treatment in DDR mutation-positive mCRPC.

This was a prospective study with well-balanced baseline clinical and demographic characteristics and a full overview of the natural history of PCa disease, but the shortcoming of an insufficient power of statistical testing should be noted, as a small number of patients with specific DDR mutations could affect the statistical analysis, especially when a tendency was observed in the results. Notwithstanding this limitation, the study demonstrated a significant predictive value of non-invasive urinary DDR testing in all stages of PCa, which, in turn, could suggest personalized therapy to obtain the best possible clinical outcomes.

## 5. Conclusions

Our data reveal the clinical value of non-invasive urine-based genetic testing in mCRPC patients for treatment individualization. DDR mutation testing along with the routine screening for familial or precocious PCa should be suggested for the selection of the most appropriate treatment strategy not only in mCRPC but also in earlier stages of PCa.

## Figures and Tables

**Figure 1 biomedicines-11-00761-f001:**
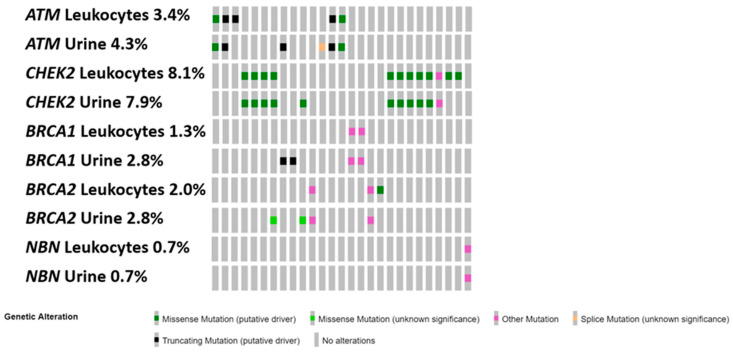
Oncoprint of selected pathogenic alterations detected in urine and leukocyte samples. Only samples with at least one detected mutation are shown. The percentages correlate to the number of mutations found in all 149 patients analysed in leukocytes and 139 in urine samples.

**Figure 2 biomedicines-11-00761-f002:**
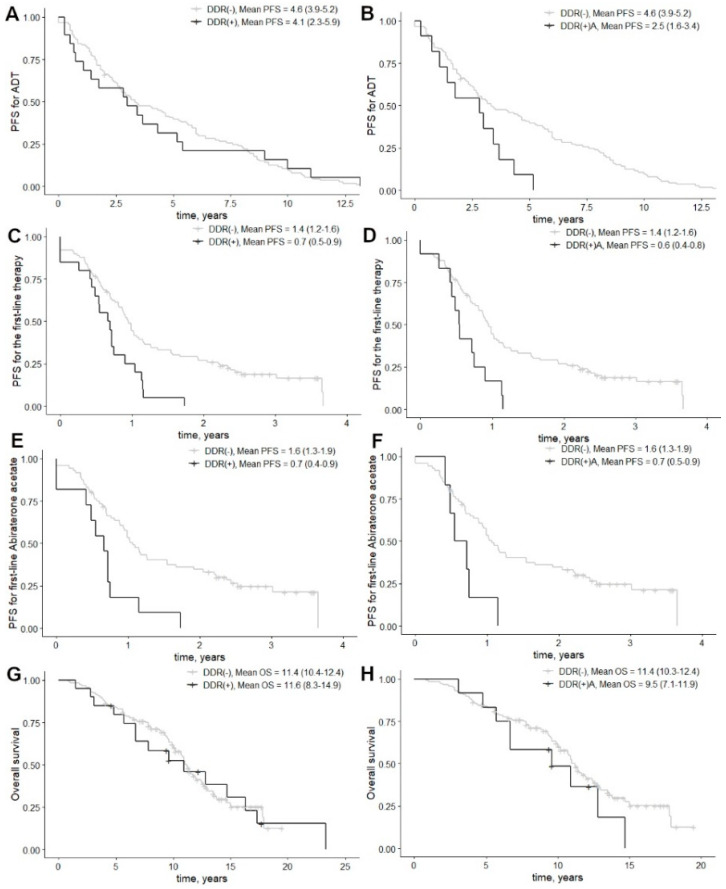
Progression-free survival for ADT in HSPC, the first-line therapy in mCRPC and overall survival. Graphs are supplied as DDR(+) vs. DDR(−) and DDR(+)A vs. DDR(−) for: ADT in HSPC (**A**,**B**), the first-line therapy (**C**,**D**), the first-line therapy with abiraterone acetate (**E**,**F**) and overall survival (**G**,**H**). The 95% confidence intervals are presented in brackets. Abbreviations: ADT—androgen deprivation therapy; HR—hazard ratio; HSPC—hormone-sensitive prostate cancer; PFS—progression-free survival.

**Table 1 biomedicines-11-00761-t001:** Molecular characteristics of variants of pathogenic or unclear clinical significance in mutation-positive patients.

**Leukocytes**	**Gene**	**Sample ID**	**Pathogenic Variants**				**Amino Acid**	**dbSNP, rs**	**Method**
			**Coding**	**Type**	**Variant**	**Exon**			
	*BRCA1*	PN088; PN179	c.4035delA	INDEL	frameshiftdel	11	*p*.Glu1346Lysfs	rs80357711	qPCR
	*BRCA2*	PN082	c.658_659delGT	INDEL	frameshiftdel	8	*p*.Val220Ilefs	rs80359604	qPCR
		PS001	c.7879A > T	SNV	missense	17	*p*.Ile2627Phe	rs80359014	NGS
		PN050	c.3847_3848delGT	INDEL	frameshiftdel	11	*p*.Val1283fs	rs80359405	NGS
	*CHEK2*	PN004; PN022; PN034; PN051; PN055; PN073; PN107; PN108; PN125; PN136; PN143	c.470T > C	SNV	missense	4	*p*.Ile157Thr	rs17879961	qPCR
		PN223	c.1100delC	INDEL	frameshiftdel	11	*p*.Thr367Metfs	rs555607708	qPCR
	*NBN*	PN132	c.657_661delACAAA	INDEL	frameshiftdel	6	*p*.Lys219fs	rs587776650	qPCR
	*ATM*	PN029; PN175	c.8122G > A	SNV	missense	55	*p*.Asp2708Asn	rs587782719	NGS
		PN044	c.1339C > T	SNV	nonsense	10	*p*.Arg447Ter	rs587779815	
		PN090; PN206	c.5932G > T	SNV	nonsense	40	*p*.Glu1978Ter	rs587779852	
	**Gene**	**Sample ID**	**Variants of Unclear Clinical Significance**				**Amino Acid**	**dbSNP, rs**	**Method**
			**Coding**	**Type**	**Variant**	**Exon**			
	*BRCA2*	PN031	c.8242_8244delGGT	INDEL	nonframeshiftdel	18	*p*.Gly2748del	–	NGS
	*NBN*	PN199	c.1445G > A	SNV	missense	11	*p*.Arg482Lys	rs775451862	
	*ATM*	PN048	c.4631A > G	SNV	missense	31	*p*.Tyr1544Cys	rs779718362	
**Urine**	**Gene**	**Sample ID**	**Pathogenic Variant**				**Amino Acid**	**dbSNP, rs**	**Method**
			**Coding**	**Type**	**Variant**	**Exon**			
	*BRCA1*	PN025	c.3268C > T	SNV	nonsense	10	*p*.Gln1090Ter	rs80357402	NGS
		PN038	c.5574G > A	SNV	nonsense	24	*p*.Trp1858Ter	rs80356914	
		PN088; PN179	c.4035delA	INDEL	frameshiftdel	11	*p*.Glu1346Lysfs	rs80357711	
	*BRCA2*	PN033	c.1532C > A	SNV	nonsense	10	*p*.Ser511Ter	rs1555281935	
		PN050; PN143	c.3847_3848delGT	INDEL	frameshiftdel	11	*p*.Val1283fs	rs80359405	
		PN082	c.658_659delGT	INDEL	frameshiftdel	8	*p*.Val220Ilefs	rs80359604	
	*CHEK2*	PN034 PN051; PN055; PN073; PN107; PN108; PN125; PN136; PN143	c.470T > C	SNV	missense	4	*p*.Ile157Thr	rs17879961	
		PN033; PN223	c.1100delC	INDEL	frameshiftdel	11	*p*.Thr367Metfs	rs555607708	
	*NBN*	PN132	c.657_661delACAAA	INDEL	frameshiftdel	6	*p*.Lys219fs	rs587776650	
	*ATM*	PN029; PN175	c.8122G > A	SNV	missense	55	*p*.Asp2708Asn	rs587782719	
		PN044	c.1339C > T	SNV	nonsense	10	*p*.Arg447Ter	rs587779815	
		PN090	c.5932G > T	SNV	nonsense	40	*p*.Glu1978Ter	rs587779852	
		PN025	c.3663G > A	SNV	nonsense	25	*p*.Trp1221Ter	rs864622490	
		PN072	c.6006 + 1G > C	SNV	unknown	40	*p*.?	rs786202016	

Abbreviations: SNV—single nucleotide variant; INDEL—insertion/deletion variant. Grey colour indicates deceased patients.

**Table 2 biomedicines-11-00761-t002:** Clinico-pathological and demographic characteristics of study cohort.

Variable	DDR(+) (*n* = 20)	DDR(−) (*n* = 117)	*p* Value
Age at PCa diagnosis, years			0.27
Median (IQR)	63.5 (57.7–71.5)	66.4 (61.5–71.1)	
Age at mCRPC diagnosis, years			0.27
Median (IQR)	63.5 (57.7–71.5)	66.4 (61.5–71.1)	
PSA level at PCa diagnosis, ng/mL			0.09
<10; n (%)	3 (15.0)	36 (30.7)	
10–20; n (%)	3 (15.0)	25 (21.4)	
>20; n (%)	14 (70.0)	53 (45.4)	
cISUP grade group			0.71
1; n (%)	7 (35.0)	48 (41.0)	
2; n (%)	4 (20.0)	18 (15.4)	
3; n (%)	2 (10.0)	14 (12.0)	
4; n (%)	5 (25.0)	16 (13.7)	
5; n (%)	1 (5.0)	12 (10.3)	
cT stage			0.62
≤T2; n (%)	6 (30.0)	43 (36.8)	
≥T3; n (%)	14 (70.0)	67 (57.3)	
Radical treatment			0.27
Radical prostatectomy; n (%)	3 (15.0)	9 (7.7)	
Radiation therapy; n (%)	9 (45.0)	42 (35.9)	
None; n (%)	8 (40.0)	66 (6.4)	
Abiraterone acetate therapy for mCRPC			0.07
First-line; n (%)	11 (55.0)	88 (75.2)	
Second-line; n (%)	8 (40.0)	26 (22.2)	
Other; n (%)	3 (15.0)	3 (2.5)	
Docetaxel therapy for mCRPC			0.77
First-line; n (%)	9 (45.0)	28 (23.9)	
Second-line; n (%)	3 (15.0)	13 (11.1)	
Other; n (%)	3 (15.0)	11 (9.4)	
Deceased			0.32
Yes; n (%)	15 (75.0)	73 (62.4)	
No; n (%)	5 (25.0)	44 (37.6)	

Abbreviations: cISUP—clinical ISUP group; cT—clinical T staging; DDR—DNA damage response; IQR—interquartile range; ISUP—International Society of Urological Pathology; n—total number of patients; mCRPC—metastatic castration-resistant prostate cancer; PCa—prostate cancer; PSA—prostate-specific antigen.

**Table 3 biomedicines-11-00761-t003:** Univariate and multivariate Cox regression analysis to assess significant factors associated with treatment and survival outcomes.

	Analysed by DDR(+)				Analysed by DDR(+)A			
	Univariate		Multivariate		Univariate		Multivariate	
	HR with 95% CI	*p* Value	HR with 95% CI	*p* Value	HR with 95% CI	*p* Value	HR with 95% CI	*p* Value
**PFS for ADT**								
DDR genetic alteration	1.04 (0.64–1.71)	0.86	1.18 (0.71–1.96)	0.53	2.11 (1.12–4.01)	**0.022**	2.17 (1.14–4.13)	**0.019**
Age	1.01 (0.98–1.04)	0.34	-		0.99 (0.97–1.03)	0.94	-	
cISUP	2.07 (1.43–3.00)	**<0.001**	2.09 (1.44–3.04)	**<0.001**	1.87 (1.28–2.74)	**0.001**	1.88 (1.28–2.75)	**0.001**
RT	1.00 (0.71–1.43)	0.97	-		1.23 (0.86–1.76)	0.26	-	
PSA0			-				-	
M vs. L	0.95 (0.58–1.56)	0.85			0.99 (0.60–1.64)	0.99		
H vs. L	1.11 (0.74–1.66)	0.62			1.14 (0.76–1.73)	0.53		
**PFS for mCRPC first-line treatment**								
DDR genetic alteration	2.17 (1.31–3.59)	**0.003**	2.22 (1.34–3.69)	**0.002**	2.47 (1.32–4.62)	**0.005**	2.53 (1.34–4.77)	**0.004**
Age	1.02 (0.99–1.05)	0.17	1.01 (0.98–1.04)	0.38	1.02 (0.99–1.05)	0.19	1.01 (0.99–1.04)	0.34
cISUP	1.26 (0.83–1.92)	0.27	-		1.22 (0.79–1.89)	0.37	-	
RT	0.77 (0.52–1.15)	0.19	0.84 (0.55–1.29)	0.43	0.77 (0.51–1.17)	0.22	-	
PSA_1_	1.001 (1.000–1.002)	**0.004**	1.001 (1.000–1.002)	**0.011**	1.001 (1.000–1.002)	**0.005**	1.001 (1.000–1.002)	**0.004**
**PFS for mCRPC first-line AA treatment**								
DDR genetic alteration	2.72 (1.39–5.33)	**0.003**	2.43 (1.23–4.83)	**0.011**	2.72 (1.14–6.50)	**0.025**	2.25 (0.92–5.49)	0.076
Age	1.03 (0.99–1.06)	0.08	1.02 (0.99–1.06)	0.24	1.03 (0.99–1.07)	0.06	1.03 (0.99–1.07)	0.18
cISUP	1.13 (0.64–2.02)	0.67	-		1.05 (0.57–1.93)	0.88	-	
RT	0.67 (0.41–1.10)	0.11	0.84 (0.48–1.46)	0.53	0.65 (0.39–1.10)	0.11	0.85 (0.48–1.49)	0.57
PSA_1_	1.001 (1.001–1.002)	**0.011**	1.001 (1.000–1.002)	**0.007**	1.001 (1.001–1.002)	**0.001**	1.001 (1.000–1.002)	**0.002**
**Overall survival**								
DDR genetic alteration	1.07 (0.60–1.90)	0.82	1.62 (0.87–3.02)	0.126	1.47 (0.73–2.95)	0.28	2.02 (0.98–4.17)	0.058
Age	1.06 (1.03–1.09)	**<0.001**	1.03 (0.99–1.06)	0.128	1.05 (1.02–1.09)	**0.004**	1.02 (0.99–1.06)	0.22
cISUP	3.15 (1.97–5.02)	**<0.001**	3.16 (1.93–5.15)	**<0.001**	2.83 (1.76–4.55)	**<0.001**	3.01 (1.83–4.94)	**<0.001**
RT	0.34 (0.21–0.54)	**<0.001**	0.39 (0.23–0.67)	**0.001**	0.39 (0.24–0.63)	**<0.001**	0.40 (0.23–0.69)	**<0.001**
PSA_1_	1.001 (1.000–1.002)	**0.022**	1.000 (0.999–1.001)	0.614	1.001 (1.000–1.002)	**0.047**	1.000 (0.99–1,001)	0.70

Abbreviations: CI—confidence interval; cISUP—clinical ISUP grade groups 3–5 vs. 1/2; mCRPC—metastatic castration-resistant prostate cancer; HR—hazard ratio; ISUP—International Society of Urological Pathology; PFS—progression-free survival; RT—radical treatment; PSA0—PSA at PCa diagnosis, ng/mL: L—<10, M—10–20, H—>20; PSA1—PSA before first-line mCRPC treatment; Numbers in bold indicate *p*-values <0.050.

## Data Availability

The data presented in this study are available on request from the corresponding author.

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
