# Peer review of "Urinary DNA as a Tool for Germline and Somatic Mutation Detection in Castration-Resistant Prostate Cancer Patients"

_biomedicines, 2023, doi:10.3390/biomedicines11030761_

Round 1
Reviewer 1 Report
In this study, the authors investigated the germline and/or somatic DNA damage response (DDR) pathway gene mutations including BRCA1, BRCA2, CHEK2, ATM, and NBN in blood and urine samples from patients with mCRPC and correlated the DDR mutations with responses to various therapeutic outcomes and survival outcomes. DDR gene mutations were assessed prospectively in DNA samples from leukocytes and urine sediments from 149 mCRPC patients using 5-gene panel target sequencing.
The data indicated that mutations detected in urine samples covered both the germline and somatic mutation in most of the DDR mutation detected samples (Figure1), confirmed the benefit of non-invasive urine-base genetic testing for timely identification of high-risk prostate cancer cases for treatment personalization.
The weakness of the study is the relatively small sample size when DDR gene mutations correspond to each specific treatment. As indicated in the manuscript,149 mCRPC patient’s cohort included 5 different treatments: Androgen deprivation therapy; First-line mCRPC therapy with abiraterone acetate; Radium-223 dichloride therapy; Taxane-based chemotherapy; Second line abiraterone acetate therapy after previous taxane-based chemotherapy (DDR mutation).
The authors need to answer the following specific questions to make the manuscript better.
1. The authors used two methods, “quantitative PCR” and “targeted next generation sequencing” to detect the mutations of DNA damage response (DDR) pathway genes, No qPCR data were shown and discussed. In page 3, line 130 “Germline DNA from DDR mutation-negative cases in qPCR screening and all urine samples was sequenced using a targeted five-gene panel”, Please indicated how many germlines DNA were DDR mutation-negative cases and were sequenced. It was confused and was not consistent with description in the results 3.1, “in a cohort of 149 mCRPC cases, the 23 patients (15.4%) were identified with germline mutations of the selected DDR genes in blood cells”.
2. In page1, Line 28, please change “next-generation sequencing” to “five-gene panel target sequencing.”
3. In page1, line 31, To make DNA damage response (DDR) pathway gene names clearer and more consistent, please change BRCA1/2 into BRCA1, BRCA2, remove c.1100delC after CHEK2.
4. In page3, line 111-112. The meaning of the word “repleted” in the sentence “The cell pellets were washed with PBS and repleted” is not clear.
5. In page3, line 115-117, “Viral RNA Mini Kit” is not consistent with “DNA concentration and purity” in the description “Urine samples were processed within 30 minutes after the samples were taken, using Viral RNA Mini Kit (Qiagen, Hilden, Germany) according to manufacturer instructions. DNA concentration and purity were determined…….”
6. In page3, line 131, it will be better if adding the name of the five genes after “using a targeted five-gene panel”.
Reviewer 2 Report
Authors should be congratulated for their work. The topic is exciting and Intriguing. The noninvasive detection of mutation in castration-resistant prostate patients is a gold standard to be achieved. I suggest improving the quality of the manuscript by adding information from this manuscript (PMID: 33958297). A major revision is required.
Round 2
Reviewer 1 Report
Based on the samples number and mutations found (7 in 133 germline DNA sample) using NGS, can author provide information about how many samples have the same mutation in table1? In addition, is there any mutation was detected by both qPCR and NGS, which should be indicated in the table1 too?
In addition, for the comment: The weakness of the study is the relatively small sample size when DDR gene mutations correspond to each specific treatment. As indicated in the manuscript,149 mCRPC patient’s cohort included 5 different treatments: Androgen deprivation therapy; First-line mCRPC therapy with abiraterone acetate; Radium-223 dichloride therapy; Taxane-based chemotherapy; Second line abiraterone acetate therapy after previous taxane-based chemotherapy (DDR mutation).
Is there any possibility that the authors could recruit more patients to make the data more solid?
Reviewer 2 Report
Authors should be congratulated for their work. They satisfactory answered to all my concerns. The manuscript is suitable for publication
Round 3
Reviewer 1 Report
The authors answered all my questions